# Comparing Mediators and Moderators of Mental Health Outcomes from the Implementation of Group Problem Management Plus (PM+) among Venezuelan Refugees and Migrants and Colombian Returnees in Northern Colombia

**DOI:** 10.3390/ijerph21050527

**Published:** 2024-04-24

**Authors:** Lucy Miller-Suchet, Natalia Camargo, Manaswi Sangraula, Diany Castellar, Jennifer Diaz, Valeria Meriño, Ana Maria Chamorro Coneo, David Chávez, Marcela Venegas, Maria Cristobal, Annie G. Bonz, Camilo Ramirez, Ana Maria Trejos Herrera, Peter Ventevogel, Adam D. Brown, Matthew Schojan, M. Claire Greene

**Affiliations:** 1Program on Forced Migration and Health, Heilbrunn Department of Population and Family Health, Mailman School of Public Health, Columbia University, New York, NY 10032, USA; mg4069@cumc.columbia.edu; 2HIAS, Silver Spring, MD 20910, USA; maria.cristobal@hias.org (M.C.); annie.bonz@hias.org (A.G.B.); matthew.schojan@hias.org (M.S.); 3Department of Psychology, Universidad del Norte, Puerto Colombia, Barranquilla 080001, Colombia; nataliacamargo@uninorte.edu.co (N.C.); jediaz@altamira.edu.co (J.D.); chamorroa@uninorte.edu.co (A.M.C.C.); atrejos@uninorte.edu.co (A.M.T.H.); 4Department of Psychiatry and Behavioral Sciences, The George Washington University, Washington, DC 20037, USA; manaswi@email.gwu.edu; 5HIAS Colombia, Barranquilla 080002, Colombia; diany.castellar@hias.org; 6HIAS Colombia, Cali 760046, Colombia; david.chavez@hias.org; 7HIAS Colombia, Bogotá 110221, Colombia; marcela.venegas@hias.org (M.V.); camilo.ramirez@hias.org (C.R.); 8Public Health Section, Division of Resilience and Solutions, United Nations High Commissioner for Refugees, CH-1211 Geneva, Switzerland; 9Trauma and Global Mental Health Laboratory, The New School for Social Research, New York, NY 10011, USA; brownad@newschool.edu; 10Department of Psychiatry, School of Medicine, New York University, New York, NY 10016, USA

**Keywords:** Colombia, forced migration, Group PM+, mediators, mental health, migrants, moderators, refugees, Venezuelans

## Abstract

Colombia hosts the largest number of refugees and migrants fleeing the humanitarian emergency in Venezuela, many of whom experience high levels of displacement-related trauma and adversity. Yet, Colombian mental health services do not meet the needs of this population. Scalable, task-sharing interventions, such as Group Problem Management Plus (Group PM+), have the potential to bridge this gap by utilizing lay workers to provide the intervention. However, the current literature lacks a comprehensive understanding of how and for whom Group PM+ is most effective. This mixed methods study utilized data from a randomized effectiveness-implementation trial to examine the mediators and moderators of Group PM+ on mental health outcomes. One hundred twenty-eight migrant and refugee women in northern Colombia participated in Group PM+ delivered by trained community members. Patterns in moderation effects showed that participants in more stable, less marginalized positions improved the most. Results from linear regression models showed that Group PM+-related skill acquisition was not a significant mediator of the association between session attendance and mental health outcomes. Participants and facilitators reported additional possible mediators and community-level moderators that warrant future research. Further studies are needed to examine mediators and moderators contributing to the effectiveness of task-shared, scalable, psychological interventions in diverse contexts.

## 1. Introduction

Over the past decade, a political and economic crisis has unfolded in Venezuela due to a variety of factors, including corruption, an authoritarian regime, and an economic crisis. The resulting lack of job opportunities, insecurity, and scarcity of basic goods has led to the mass migration of millions of individuals out of the country [1]. Neighboring Colombia hosts the largest number of Venezuelan migrants and refugees of any country [2]. This mass influx of migrants has strained the public health and social services system in Colombia, which lacks the capacity to fully address the health and mental health care needs of the millions of individuals who seek refuge in Colombia [3]. Qualitative accounts of Venezuelan migrants and refugees in Colombia highlight marked difficulties in access to treatment for health and, more specifically, mental health conditions [4,5]. Evidence-based task-sharing interventions involve delegating tasks typically performed by mental health specialists to non-specialists with adequate training and supervision [6,7]. Task sharing can be used to enable lay personnel to deliver short-term support for common mental health conditions to bridge the gap in access to mental health services in communities with limited resources [8,9], including war-affected, forcibly displaced, migrant, refugee, and host community populations [6,10]. A growing body of evidence shows that task-sharing interventions are effective in reducing distress and improving access to mental health and psychosocial support (MHPSS) services [11,12,13,14,15,16]. International humanitarian guidance from the World Health Organization (WHO), the United Nations High Commissioner for Refugees (UNHCR), the Inter-agency Standing Committee (IASC), and others recommend the provision of community-based MHPSS interventions, including their delivery through scalable implementation strategies, such as task sharing [17,18,19,20,21].

Problem Management Plus (PM+) is a task-sharing, MHPSS intervention developed by the WHO, based on psychoeducation and the acquisition of problem-solving skills and cognitive-behavioral strategies for stress management and strengthening social support [22]. Individual and group formats of PM+ have effectively reduced psychological distress (i.e., mental health-related symptoms such as depression, anxiety, and stress) among conflict-affected populations in countries such as Nepal, Pakistan, Turkey, and Jordan [23,24,25,26,27,28].

Even though research over the last decade has shown that there is a direct relationship between social determinants of health (such as marginalized identities, employment opportunities, and exposure to adversity) and mental health issues that individuals face [29], few studies of community-based MHPSS interventions have examined moderators (i.e., which groups benefit most) and mediators (i.e., how the intervention works) of intervention effects, including for Group Problem Management Plus (Group PM+). Moderators examine the characteristics and specific context of a population and how they correlate with different mental health trajectories. Mediators provide insight into the process of symptom reduction and how the intervention works to improve well-being [30]. One study in Nepal examined the utilization of PM+ psychosocial skills, the hypothesized mediator, via the Reducing Tension Checklist (RTC), and found that the use of psychosocial skills was greater in the Group PM+ arm immediately after receiving the intervention when compared to standard care [28]. Yet, there are few studies examining moderators and no additional studies of mediators of individual or Group PM+ and many other community-based, task-shared, MHPSS interventions. The existing studies that examine task-sharing MHPSS interventions have highlighted the need for future research into which groups are most likely to improve from such interventions [16].

Evidence and findings generated by this study aim to identify whether certain factors contribute to positive Group PM+ outcomes and provide further evidence into psychosocial skill acquisition as a mediator of this intervention. Additionally, as the WHO, IASC, UNHCR, and other multilateral organizations are recommending the implementation of task-sharing interventions, it is important to shed light on key factors that affect the impact and success of these interventions, ensuring that rollout is as effective as possible across the world.

The objective of this study was to conduct a secondary analysis of a randomized effectiveness-implementation trial to explore moderators and mediators of Group PM+ in Barranquilla and Soledad, Colombia, using qualitative and quantitative data. Detailed study procedures are found in the protocol paper [31] and the original aims and findings of this study are described and analyzed in the parent study [31]. This study analyzed problem management skills obtained by participants 1 week after receiving the intervention (i.e., endline) as a mediating variable between Group PM+ session attendance and participant mental health outcomes. This study also examined the following participant baseline characteristics as potential moderators of Group PM+ and mental health outcomes: type of identification, education, nationality, ethnicity, employment, head of household status, previous mental health services used, past-year history of gender-based violence (GBV), and study site. We analyzed qualitative data to triangulate findings and gain further insights into moderators and mechanisms of action of Group PM+ from the point of view of the participating women and the facilitators. We chose these potential moderators based on the social determinants of health framework in order to better understand how social determinants affect the relationship between this task-sharing intervention and mental health outcomes [32].

## 2. Materials and Methods

### 2.1. Setting

The Venezuelan migration crisis reached its peak in 2017–2018 [33]. The flow of people included Colombian returnees and Venezuelans migrating to or transiting through Colombia. Many of these migrants and refugees settled in the northern and central areas of the country [33]. Approximately 2.8 million Venezuelans live in Colombia, including 149,165 in the city of Barranquilla and 32,068 in the municipality of Soledad [34].

HIAS Colombia, a branch of HIAS, is a non-governmental organization that works with Venezuelan refugees and migrants, Colombian returnees, internally displaced persons, and host communities. Since its creation in 2019, HIAS Colombia has provided MHPSS and other services to communities to promote well-being and support them in rebuilding their lives in Colombia and throughout their migration journey. This research was a collaboration between The New School for Social Research, Columbia University, Universidad del Norte, and UNHCR, as well as several additional institutions that contributed to the parent study. Researchers from The New School, Columbia, Universidad del Norte, and UNHCR designed the research project, served as principal investigators, and provided oversight throughout implementation. Individuals from additional institutions provided quarterly guidance on implementation via a scientific advisory committee. HIAS Colombia was the implementing partner and conducted the research in communities in the city of Barranquilla and the municipality of Soledad. These communities included the neighborhoods of Primero de Mayo, Villa Caracas, and Santa María. These communities were chosen due to high influxes of Venezuelan migrants and Colombian returnees, as well as high levels of insecurity. They are also central locations for individuals in neighboring communities with high levels of migrants [35,36]. We collected data and conducted the Group PM+ sessions in community centers and public schools in these neighborhoods. HIAS Colombia provided MHPSS and other support services throughout the duration of this project.

### 2.2. Participants and Procedures

#### 2.2.1. Study Design Overview

This study was nested within a parent study, a type II hybrid effectiveness-implementation trial [37], of which the primary aim was to compare training and supervision strategies for Group PM+. In the parent study, adult women residing in one of the three study communities who reported elevated levels of distress and functional impairment at baseline were enrolled and randomized to receive the intervention, Group PM+, under two conditions. In the first condition, lay facilitators were trained and supervised by specialized psychologists to deliver Group PM+ to eligible women (specialized technical support arm). In the second condition, the group of lay facilitators from phase one were trained as trainers and subsequently trained and supervised a new cadre of lay facilitators to deliver Group PM+ (non-specialized technical support arm). As shown in Figure 1, participants allocated to the specialized support arm were immediately assigned to an intervention group and began Group PM+. Participants allocated to the non-specialized support arm initially entered a waitlist period and began Group PM+ approximately three months after randomization. This crossover design with a waitlist phase was needed because the trainers for the non-specialized condition were initially trained as facilitators and delivered Group PM+ as part of the specialized technical support arm before they were trained as trainers and served as trainers/supervisors in the non-specialized technical support arm.

In this secondary analysis, we used data from the parent study to explore mediators and moderators of Group PM+ [37]. We used data from all study participants enrolled in the parent study, as well as data collected at all time points. The study design remained the same as that of the parent study; however, the study objectives more specifically focused on exploring mediators and moderators of observed effects of the intervention on mental health outcomes. Study design details (e.g., recruitment, outcome measures, etc.) are consistent with those of the parent study and are described below.

#### 2.2.2. Study Sample and Recruitment

The sample consisted of women over 18 years of age who planned to live in the city of Barranquilla or the municipality of Soledad for at least three months after screening. Participants with moderate functional impairment (WHO Disability Assessment Schedule 2.0, WHODAS > 16) and psychological distress (General Health Questionnaire 12, GHQ-12 > 2) at baseline were enrolled in the study [38,39]. Participants at risk of suicide and those who had cognitive impairment were excluded from study participation. Participants with suicidal intent or ideation were referred to trained psychologists at HIAS.

We extended invitations to community members registered in a contact list supplied by HIAS Colombia and individuals recommended by Group PM+ facilitators and other community leaders. Participants were contacted by telephone and informed about the aims of the study and participation criteria of the project before being asked for verbal consent to participate. Then, a pre-screening process was carried out, soliciting demographic information such as gender identity, length of stay, GHQ-12, and the WHODAS. Women who met the inclusion criteria and gave written informed consent were asked to complete a baseline assessment and randomized to a study condition. Participants who were actively receiving Group PM+ were asked to complete an endline assessment one week after intervention completion. All participants were asked to complete a follow-up assessment three months after enrollment. All data were collected by HIAS staff and researchers from the Universidad del Norte, with support from The New School for Social Research. The instruments used in the research were implemented by HIAS and the Universidad del Norte. Some of the assessments were conducted by telephone due to COVID-19, and some were conducted face to face, depending on the ability of the assessor and the respondent. Quantitative data were collected using KoBoToolbox v2021.2.4., an instrument for data collection, management, and data visualization [40]. One focus group discussion (FGD) was held in each of the three communities with participants after the completion of Group PM+ to collect qualitative data. Research assistants selected participants for the FGDs based on their interest, availability, and whether they had attended at least one session of the intervention. These FGDs were part of a process evaluation and aimed to gather information from participants about their experiences with Group PM+ and how the intervention affected their mental health, subsequently impacting their family and their communities.

#### 2.2.3. Intervention and Implementation Strategies

Group PM+ is a transdiagnostic, brief, five-session intervention that can be delivered by both specialists and non-specialists to address mental health problems such as depression, anxiety, and stress in communities affected by adversity [41]. As part of the specialized technical support arm, sessions were delivered by lay facilitators who were trained and supervised by psychologists to deliver Group PM+. Women who were active in their communities, interested in participating, and had time available to deliver the sessions were invited to be facilitators. After delivering Group PM+ to their designated participants, a subset of these facilitators participated in a training of trainers and subsequently became trainers/supervisors themselves. They then trained other new facilitators who would be responsible for delivering the program sessions as part of the non-specialized technical support arm. The training of trainers took place at the Universidad del Norte, where the lay supervisors were taught theoretical components of how to train Group PM+ facilitators and practiced effectively delivering facilitator training and supervision. These lay supervisors then trained and supervised the new facilitators as part of the non-specialized technical support arm.

#### 2.2.4. Measures

Demographic measures such as age, level of education, employment status, and legal status of the participants were collected during baseline assessments. Legal status was measured via two proxy indicators, nationality and the type of identification (ID) they possessed [foreign ID, Colombian ID, a temporary permit of permanence (PTP), or special permit of permanence (PEP)]. Participants were each assigned to a site, Villa Caracas, Santa Maria, or Primero de Mayo, according to their proximity to each location. Therefore, the variable “site” was utilized as a proxy for community characteristics that each participant experienced.

The following instruments were used to assess participant mental health outcomes: the Patient Health Questionnaire (PHQ-9) [42], which consists of a 10-item Likert scale designed to measure participants’ levels of depression and distress. The Posttraumatic Stress Symptoms Checklist (PTSD Checklist for DSM-5, PCL-5) was also utilized [43]. The PCL-5 is a 20-item self-report measure designed to identify post-traumatic stress symptoms according to the DSM-5. The Psychological Outcomes Profiles (PSYCHLOPS) [44], a 4-item instrument in which participants self-report levels of well-being and distress, was also utilized. Finally, the RTC is a 10-item instrument that evaluates the acquisition of Group PM+ coping skills such as “stress management through deep breathing, problem-solving, behavioral activation, and seeking social support” [28]. Additional information about quantitative outcome measures is included in the study protocol [37].

Qualitative FGD interview guides for participants and facilitators included questions related to coping skills and strategies learned in PM+, perceived benefits and impact of the intervention, and barriers to session participation. Questions included the following: “what did you like about PM+?”, “how did the program impact you and your family?”, and “what skills and strategies learned in PM+ were the most helpful for you?”.

### 2.3. Analysis

We undertook quantitative and qualitative data analysis with the goal of triangulating data from both of these methods to gain further insight into moderators and the intervention’s mechanisms of action. For the quantitative analysis, we first described the distribution of demographic and migration characteristics between study conditions at baseline. We compared these distributions using *t*-tests for continuous variables and chi-squared analyses for categorical variables. To examine possible moderators of Group PM+, we compared the change in mental health outcomes from baseline to the 3-month follow-up between participants receiving Group PM+ under the specialized technical support study condition to those who were allocated to the non-specialized technical support study condition and were currently in the waitlist period. We constructed linear regression models examining the association between Group PM+ (vs. waitlist) on the change in mental health outcome from baseline to the 3-month follow-up, controlling for age and stratified at each level of the hypothesized moderator (study site, education, ethnicity, employment, head of household, family composition, identification type, baseline mental health, prior use of mental health services, and past-year GBV). We adopted a social determinants of health framework when determining which moderators to include in order to research if and how social determinants affect the relationship between Group PM+ and themental health outcomes examined [32]. We also constructed models that included an interaction between study condition and the moderator to assess whether observed differences across strata were statistically significant (*p* < 0.05). To evaluate whether the use of Group PM+ skills mediated the relationship between Group PM+ attendance and mental health outcomes, we used Baron and Kenny’s model [45] for evaluating mediation (see Figure 2). We first estimated the total effect of Group PM+ attendance (any attendance and number of sessions attended) on mental health outcomes at 3 months post-intervention in both study conditions. We then evaluated whether Group PM+ attendance predicted use of Group PM+ skills at endline, followed by evaluating whether Group PM+ skill use predicted mental health outcomes at 3-month follow-up. Finally, we constructed a linear regression model of mental health outcomes at 3-months as a function of both Group PM+ attendance and skill use. All models controlled for age at baseline.

For the qualitative analysis, FGDs were undertaken with Group PM+ participants and grouped by community. The audio from each focus group was recorded and thereafter transcribed using NVivo software v14 [46]. Transcripts were analyzed using a thematic analysis approach [47]. Two independent researchers coded each of the interviewers in Dedoose v9.0.107 [48]. Differences were discussed and common themes and codes were identified and described via discussions amongst the qualitative analysis team. Five themes and eighteen sub-themes were extracted for analysis within the parent study. Five sub-themes were identified as relevant to explanations for possible mediators and moderators. Once coding was undertaken in Dedoose, memos were developed that detailed the study codes and themes.

### 2.4. Ethics

All participants provided written informed consent prior to enrollment. All procedures were reviewed and approved by the IRB at the Universidad del Norte (#237). Approval for secondary analysis was obtained at Columbia University (AAAU3933).

## 3. Results

### 3.1. Description of the Sample at Baseline

Overall, we aimed to enroll 128 participants, but ultimately randomized 127 due to 1 participant declining participation after completing a baseline assessment and before randomization. The 127 participants were randomized to eight groups with approximately eight participants in each. More than half of the participants were recruited from the community of Villa Caracas (67.3%), followed by Primero de Mayo (21.5%) and Santa Maria (11.2%). At baseline, participants were 33.3 years old, on average (SD = 10.7). Most participants identified as Mestizo or no ethnicity (90%), had obtained more than a primary school education (87.4%), were the head of their household (81.1%), maintained identification from another country [i.e., not Colombian identification (59.8%)], had not experienced GBV in the past year (80.2%), and had no prior history of utilizing MHPSS services (80.2%). Approximately half of the participants (48.4%) were unemployed, students, volunteers, conducted household labor, or other. The remaining participants undertook salaried, formal, or self-employed work (27.8%) or informal work (23.8%). All baseline characteristics described can be found in Table 1.

### 3.2. Baseline Characteristics as Moderators of the Relationship between Group PM+ and Mental Health Outcomes

Our results showed that most moderators examined, when controlling for age, did not significantly modify the association between Group PM+ (vs. waitlist) and mental health outcomes 3- months post-intervention in models as evaluated by the significance of the interaction between study condition and the moderator.

We observed patterns that may indicate potential moderators in stratified analyses that, while not statistically significant, warrant further investigation in a fully powered analysis (Table 2). Group PM+ appeared to be more strongly associated with reductions in mental health symptoms for participants with more than a primary school education relative to those with less than a primary school education; participants who identified as Colombian or Colombian-Venezuelan relative to Venezuelan; participants who identified as Mestizo or no ethnicity relative to Afro-Caribbean, Indigenous, or other ethnic minorities; participants who had Colombian identification relative to PTP/PEP; participants who engaged in salaried/formal work or were self-employed relative to those within the formal work sector; participants who reported having experienced GBV during the past year relative to those who had not; participants who were recruited from the Primero de Mayo neighborhood; and participants who participated in at least one session of Group PM+ relative to those who did not attend any sessions. All coefficients from the stratified analyses are provided in Table 2.

Participants and facilitators noted several contextual factors that modified the implementation and, potentially, the effectiveness of Group PM+. FGDs revealed that study communities varied in terms of insecurity, how affected they were by severe weather and environmental conditions, geography and the transportation options or requirements to attend sessions, as well as socioeconomic conditions that contributed to participant’s ability to meet other basic needs—especially related to health, childcare, and work opportunities—that sometimes interfered with their ability to attend sessions.

*“The time [is a big challenge], because really what you navigate is tremendous, because sometimes you have to say that you already have some commitments already practically. And sometimes you stop doing other things to continue with Group PM+”*.—FGD, post-implementation, participant, Santa Maria

*“The first day I did not attend because it rained that day and the baby was sick. I took her to the doctor and to get medicine, but then it started to rain and I couldn’t come that day”*.—FGD, post-implementation, participant, Villa Caracas

Descriptions of these contextual factors varied across communities, suggesting that these characteristics may also influence Group PM+ engagement and outcomes.

### 3.3. Group PM+ Skill Use as a Mediator of the Relationship between Group PM+ Attendance and Mental Health Outcomes

Approximately 65% of participants (*p* < 0.001) attended one or more sessions and the average number of sessions attended was 2.31 out of a total of 5 sessions (SD = 2.00, *p* < 0.001). When examining RTC as a mediator of attendance, we observed that individuals who attended more sessions displayed significantly higher levels of PM+-related skill use at endline (B = 1.19, 95% CI: 0.27, 2.10). Attending at least one session was also associated with an increase in skill use at endline, but this was not statistically significant (B = 3.80, 95% CI: −0.65, 8.25). Group PM+ skill use at endline was not significantly associated with any of the mental health outcomes at 3-month follow-up. Furthermore, we did not observe a total effect of any attendance or number of sessions attended on mental health outcomes at three-month follow-up (Table 3).

Results from qualitative focus groups revealed additional potential mechanisms of intervention effects. Participants and facilitators noted that participants developed additional coping skills and strategies via Group PM+ that extended beyond the use of the main four PM+ strategies assessed by the RTC. These additional skills and strategies included communicating more confidently, expressing and understanding one’s challenges better, and developing patience and reducing anger. Many individuals explained that they learned communication skills and were more comfortable, less fearful, and more expressive when communicating with others. A facilitator noted this change in her participants:

*“There were participants where they fought with their husbands and fought constantly and one of them commented in the sessions “we don’t even fight anymore” and what we do is talk, let’s take at least an hour in the afternoon to talk. And when she arrived at the last session, she told us herself, she said “I’m okay with it, I’m satisfied”. And that is what we achieved, that there are life changes”*.—FGD, post-implementation, facilitator

Participants also stated that they were better equipped to express and understand their challenges as a result of Group PM+, which helped alleviate some of the mental health issues they were facing. Participants reported a clearer understanding of how to better express their feelings, which gave them the ability to better solve challenges that arose. Additionally, participants explained that they were able to better understand that their problems might not have direct solutions, which led participants to let go of the distress associated with the problem.

*“This helped me a lot because one [writes down] the difficulties that one cannot solve. And even then you don’t get [an answer] and solve it”*.—FGD, post-implementation, participant, Primero de Mayo

Participants also finished the intervention feeling as though they had developed patience, particularly in dealing with their children, families, and spouses. One participant expressed that before the intervention, she used to have difficulty controlling her anger and would find herself physically punishing her son when she ran out of patience. However, now she has learned how to be more patient and communicate better with her son, improving their relationship and her feelings towards the situation.

*“So in this part I learned to deal with these things, it has helped me a lot with my son, because I had no patience for him and I was the type of person that if he didn’t listen to me the first time I spoke to him, I was hitting him. So [the intervention] helped me a lot… For example, I now know how to talk to him, this is so much, it has helped me and I have other knowledge and now I am more comfortable, I can talk to him more, he understands more and it’s better”*.—FGD, post-implementation, participant, Villa Caracas

Along with suggesting additional potential mechanisms of change beyond those measured in the RTC, some participants specifically discussed the usefulness of the PM+ strategies. They noted that some PM+ strategies were more useful than others. A participant shared how the behavioral activation strategy helped her break free from a cycle of inactivity:

*“I had fallen into a cycle of inactivity in which I couldn’t get out, I stayed at the house, I was just fighting with the children, with my partner, I didn’t get dressed up, I was always frozen like I was crazy and everything was a stress, a fight, I was swearing and this has helped me to control myself a little bit now. I mean, I get dressed up, I go out with my partner, I go out with the kids, we go to the park, we eat, and now I feel more, better, freer, more active, calmer, I leave the house, because sometimes the house becomes a monotony, the house, the house, the house, the house. So you can get out of that monotony”*.—FGD, post-implementation, participant, Villa Caracas

## 4. Discussion

This exploratory analysis has produced findings that may serve to guide future MHPSS task-sharing studies and implementation. Due to small sample sizes, we can only draw preliminary observations and generate hypotheses based on patterns found in the results. In general, we observed the impact of a number of social determinants on participant-level mental health outcomes; individuals in more stable, less marginalized positions improved more than participants with less stable, more marginalized identities. This pattern was present in the moderation analyses and suggests less benefit is derived from Group PM+ among people with less secure legal status and identification, lower levels of education, foreign nationality, minority ethnicity, and less stable employment. However, further research with larger sample sizes is needed to evaluate and confirm these preliminary hypotheses and patterns.

Two possible explanations for this pattern are that individuals in less stable or marginalized groups may have experienced more difficulties in attending sessions due to systemic barriers in seeking mental health support. This was supported by qualitative data collected from participants and providers, many of whom stated that individuals struggled to participate due to competing priorities such as caring for their children or job opportunities. Individuals in less stable, more marginalized positions likely encountered more systemic barriers to participation. This possible difficulty in attending sessions may have led individuals to receive a lower dose of the intervention and derive less benefit. Another possible explanation is that the problems that individuals in more marginalized positions encounter are too profound for a short-term intervention. Minorities and marginalized individuals globally experience higher levels of stress, adverse childhood experiences (ACEs), racism, insecurity, and additional challenges related to social context [49,50], leading them to require longer or more extensive mental health care than a 5-session intervention may provide.

Two exceptions to this pattern in moderators were observed. First, individuals who experienced GBV in the past year appeared to improve more across all outcome measures than individuals who had not experienced GBV in the past year. A potential explanation for this pattern is that individuals in these communities may not have previously been in spaces where others disclosed information about their experience with GBV, leaving them to feel alone with their experiences. Previous studies have noted the positive effects and therapeutic benefits of disclosing sexual abuse and other types of GBV when met with positive reactions [51,52,53]. By providing a space where recent GBV survivors could connect with others who had similar experiences, this intervention may have allowed individuals to experience these benefits and heal. Second, individuals who participated in Group PM+ in the site of Primero de Mayo seemed to report the largest improvements in all three outcome measures. Primero de Mayo is the most dangerous of the three neighborhoods in this study, according to the announcement of the mayor of the municipality of Soledad, who noted that the neighborhood is considered a dangerous area due to high rates of robberies and murders [54]. Yet, individuals in this community improved more than individuals in the more stable communities, Santa Maria and Villa Caracas. While individuals in Santa Maria and Villa Caracas experience violence, insecurity, and limited access to MHPSS programs, services in Primero de Mayo are significantly more limited. Areas with higher levels of violence are often associated with greater feelings of isolation, fear, vigilance, and lack of social connection, particularly for women [55,56]. For example, women in an informal migrant camp in Mexico experienced negative mental health effects of living in an area with high levels of violence, leading to significant impacts on mental health from constant vigilance, feelings of isolation, and more [55]. As the more dangerous neighborhood of the three intervention sites, individuals in Primero de Mayo may have felt less secure traveling to receive social services that were not located in their community. Because of high adversity, greater need, and fewer social services aside from PM+, this intervention delivered within the community may have helped fill a critical void in Primero de Mayo. To this end, participants in more adverse neighborhoods noted that Group PM+ was especially necessary in their neighborhoods because they did not have programs similar to this nor access to more advanced psychological care. Another possible explanation for different changes in outcomes across communities is between group differences related to participant group cohesion, PM+ facilitator’s competency and technical skills, and fidelity to the intervention.

Contrary to our hypotheses, while attending Group PM+ did increase the use of PM+ skills, as measured by the RTC, the use of PM+ skills did not appear to influence changes in mental health outcomes at the 3-month assessment time point. We also did not find significant associations between attendance and changes in mental health outcomes immediately following the intervention. Our analysis did not find strong evidence of differential mental health outcomes by greater session attendance. When accounting for the mediator, RTC, there were minimal changes in outcome measures, suggesting that the small changes in symptoms at endline were likely not related to PM+ skill use. In qualitative interviews, participants and facilitators supported these findings, noting that skills, such as the ability to increase confidence and maintain patience, which were not skills measured by the RTC, were key in improving mental health outcomes and increasing motivation to attend sessions. This corroborates the findings from the parent study, which revealed that most of the effects of Group PM+ were observed at endline with strong attenuations observed three months post-intervention [37]. As such, the self-report nature of the RTC may not have fully captured the potential benefits of skill acquisition and future studies would benefit from examining skill learning in other ways.

Upon comparing these results to studies of mediators and moderators of similar interventions, our results were different. Our results diverged from those generated by the implementation of Group PM+ among women in disaster-prone regions of Nepal, which found that 31 percent of the treatment effect of reducing psychological distress at 3-month follow-up was mediated by participants’ use of PM+-related therapeutic strategies, even though mediation effects were limited [28]. Due to the novelty of our analysis, there are few comparable studies in the region that have examined similar moderators of MHPSS interventions. However, a systematic review and meta-analysis of PM+ (group and individual) and its digital version, Step by Step (SbS), found that older individuals experienced less positive outcomes and participants who experienced longer intervention durations had smaller improvements at follow-up [13]. Due to the limited number of similar studies and the variation between existing studies, additional research is needed to understand the mechanisms of action of Group PM+ and session attendance and to identify who benefits most from these interventions.

There are several limitations of this study that must be considered when interpreting the results. First and foremost, the small sample size precluded us from conducting a fully powered analysis of mediators and moderators of Group PM+ and intervention attendance. Thus, the results of this analysis had large confidence intervals, and findings should be considered exploratory and confirmed through future research. Additionally, this is a secondary analysis of data from a study that was not specifically designed to answer the research questions evaluated here. To this end, we did not note the baseline characteristics of individuals who participated in the FGDs, limiting our ability to triangulate the results of the moderation analysis via qualitative data. Moreover, the RTC as a mediator may not sufficiently capture all avenues of mediation, as participants and facilitators identified additional avenues for mental health improvements. Finally, the crossover study design in the parent trial may have resulted in bias due to differential completion rates, issues with temporal confounding, and added sources of variation that further reduced precision.

In spite of these limitations, this study provides important insights into Group PM+, as well as future research and practice. This study identifies potential moderators of Group PM+ and may serve to generate hypotheses about which groups potentially benefit most from this intervention. These findings are particularly important, as interventions are considered for mobile populations and groups affected by humanitarian emergencies, contributing to a body of evidence on how to best implement community-based, task-sharing MHPSS interventions, as they are becoming more common globally. This study sheds light on the importance of understanding the relationship between modifying characteristics, both included in this study and other potential moderators, and outcomes when providing Group PM+ and other MHPSS interventions, particularly in humanitarian settings where limited resources must be allocated to diverse communities. Future studies on moderators of Group PM+ are needed to determine more definitively which groups would benefit from the intervention. Additionally, future studies would benefit from collecting baseline characteristics of qualitative interviewees in order to adequately triangulate findings, as has been carried out in prior task-sharing interventions [57]. This study also begins to explore the mechanisms of action of Group PM+ and RTC as a mediator of these mechanisms. Future research on the relationship between Group PM+ skill acquisition and MHPSS outcomes would be useful, as would the modification or expansion of RTC as a mediator and the exploration of additional potential mediators (e.g., social connectedness and support). Exploring other potential mediators through theory of change workshops and qualitative studies and including measures of these hypothesized mechanisms in future studies would enable the identification and confirmation of other key mediation pathways. Future studies exploring a combined treatment model that addresses key social determinants of health in conjunction with Group PM+ or another MHPSS intervention are needed.

## 5. Conclusions

This study explored the mediators and moderators of an MHPSS intervention, Group PM+, for refugee, migrant, and host community women in northern Colombia. We found that the hypothesized mediator, RTC skill acquisition, did not show significant mediation effects. The moderation analyses showed that individuals in more stable, less marginalized positions improved more than participants with less stable, more marginalized identities. Future fully powered research on Group PM+ and other community-based mental health interventions should include analyses of mediators and moderators to fully understand mechanisms of action and which groups would most benefit from this intervention.

## Figures and Tables

**Figure 1 ijerph-21-00527-f001:**
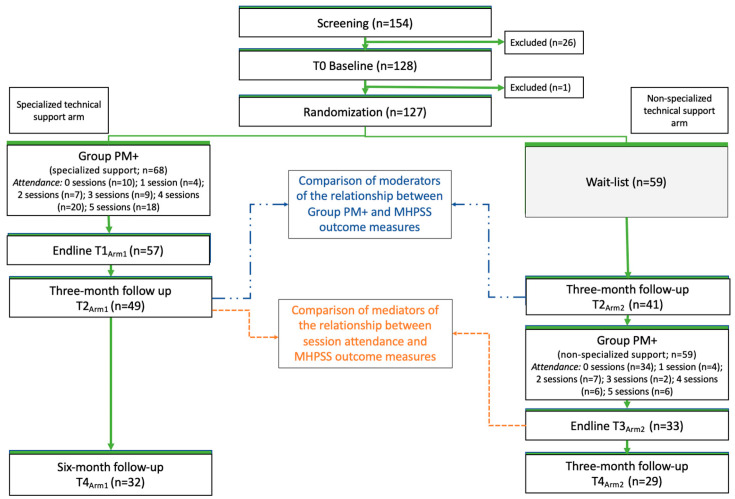
Study design and participant flow diagram.

**Figure 2 ijerph-21-00527-f002:**
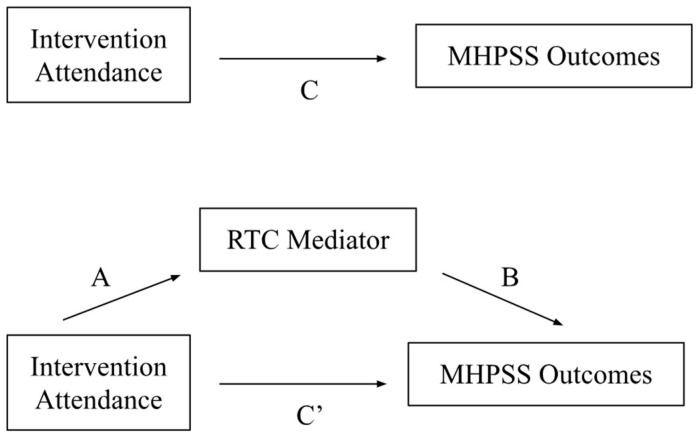
Mediation model diagram. C: total effect, C’: direct effect, A and B: indirect effect.

**Table 1 ijerph-21-00527-t001:** Baseline demographic and migration characteristics of the sample.

Characteristic at Baseline	Full Sample (*n* = 128) ^1^	Non-Specialized Technical Support (*n* = 59)	Specialized Technical Support (*n* = 68)	Significance Test Result
	Mean (sd) or *n* (%)	Mean (sd) or *n* (%)	Mean (sd) or *n* (%)	*p*-Value
Age	33.26 (10.65)	35.29 (12.06)	31.38 (9.00)	*p* = 0.039
Identification Type				*p* = 0.204
Colombian ID	21 (16.5%)	13 (22.0%)	8 (11.8%)	
Foreign ID	76 (59.8%)	35 (59.3%)	41 (60.3%)	
PEP	30 (23.6%)	11 (18.6%)	19 (27.9%)	
Education				*p* = 0.442
Primary school education only or less	16 (12.5%)	6 (10.2%)	10 (14.7%)	
Greater than primary school education	112 (87.5%)	53 (89.8%)	58 (85.3%)	
Nationality				*p* = 0.317
Colombian or Colombian- Venezuelan	21 (16.8%)	9 (13.6%)	12 (20.3%)	
Venezuelan	104 (83.2%)	57 (86.4%)	47 (79.7%)	
Ethnicity				*p* = 0.127
None	108 (90.0%)	52 (94.5%)	56 (86.2%)	
Other, Indigenous, Afro- Caribbean	12 (10.0%)	3 (5.5%)	9 (13.8%)	
Employment				*p* = 0.028
Unemployed, household, student, volunteer, other	61 (48.4%)	36 (61.0%)	25 (37.3%)	
Salaried, formal work, or self-employed	35 (27.8%)	13 (22.0%)	22 (32.8%)	
Informal work	30 (23.8%)	10 (16.9%)	20 (29.9%)	
Head of Household				*p* = 0.946
Yes	103 (81.1%)	48 (81.4%)	55 (80.9%)	
No	24 (18.9%)	11 (18.6%)	13 (19.1%)	
Prior mental health services utilized, past year				*p* = 0.820
Yes	25 (19.8%)	11 (19.0%)	14 (20.6%)	
No	101 (80.2%)	47 (81.0%)	54 (79.4%)	
Gender-based violence, past year				*p* = 0.382
Yes	24 (19.8%)	9 (16.4%)	15 (22.7%)	
No	97 (80.2%)	46 (83.6%)	51 (77.3%)	
Site				*p* = 0.941
Villa Caracas	72 (67.3%)	33 (67.3%)	39 (67.2%)	
Santa Maria	12 (11.2%)	5 (10.2%)	7 (12.1%)	
Primero de Mayo	23 (21.5%)	11 (22.4%)	12 (20.7%)	

^1^ One participant was not randomized to a group but completed a baseline assessment. These data are therefore only included in the full sample statistics.

**Table 2 ijerph-21-00527-t002:** Moderators of intervention effects.

	Between Group Difference in Change in Outcomes from T1 to T3 (Controlling for Age)		
	Depressive Symptoms	PTSD Symptoms	Self-Defined Problems
ID Type			
Colombian ID	−3.48 (−8.40, 1.44)	−6.41 (−19.00, 6.19)	−1.66 (−10.90, 7.59)
Foreign ID	−0.40 (−4.37, 3.56)	−4.16 (−14.95, 6.63)	0.28 (−3.49, 4.06)
PTP/PEP	4.22 (−2.84, 11.28)	7.09 (−8.85, 23.02)	1.69 (−3.81, 7.20)
Education			
Primary school or less	4.62 (−3.27, 12.51)	3.76 (−14.50, 22.01)	−0.46 (−9.68, 8.76)
More than primary school	−0.79 (−3.88, 2.29)	−4.09 (−12.16, 3.98)	0.00 (−3.03, 3.04)
Nationality			
Colombian or Colombian-Venezuelan	−0.71 (−9.12, 7.71)	−10.49 (−27.49, 6.51)	2.01 (−4.81, 8.84)
Venezuelan	0.73 (−2.24, 3.70)	−0.63 (−8.94, 7.67)	−0.46 (−3.66, 2.74)
Ethnicity			
Mestizo/None	−0.85 (−3.77, 2.07)	−3.74 (−11.71, 4.23)	−0.44 (−3.39, 2.49)
Afro/Indigenous/Other	8.02 (−5.01, 21.05)	8.92 (−13.86, 31.71)	2.33 (−15.32, 19.97)
Employment			
Unemployed/Student/No Income	0.66 (−3.75, 5.08)	−6.12 (−16.83, 4.57)	−0.35 (−4.70, 4.01)
Informal Work	4.36 (−2.76, 11.48)	−1.42 (−20.48, 17.64)	3.07 (−2.70, 8.85)
Salaried/Formal Work or Self-employed	−4.32 (−9.40, 0.77)	−6.88 (−20.74, 6.98)	−3.53 (−9.35, 2.28)
Head of household			
No	3.68 (−3.84, 11.21)	−7.19 (−21.55, 7.17)	3.49 (−3.04, 10.02)
Yes	−0.96 (−3.96, 2.04)	−1.21 (−9.51, 7.08)	−1.21 (−4.30, 1.89)
Prior MHPSS service use			
No	0.21 (−2.90, 3.32)	−5.01 (−13.09, 3.07)	−1.98 (−4.99, 1.02)
Yes	−3.13 (−10.91, 4.66)	−1.46 (−21.29, 18.36)	3.81 (−4.26, 11.88)
Past-year GBV			
No	0.56 (−2.45, 3.57)	0.35 (−7.89, 8.59)	−0.37 (−3.62, 2.89)
Yes	−2.19 (−11.00, 6.62)	−11.87 (−31.93, 8.19)	−1.93 (−8.67, 4.81)
Community			
Villa Caracas	1.23 (−2.19, 4.65)	−1.41 (−9.24, 6.42)	−0.84 (−4.39, 2.71)
Santa Maria	−2.02 (−16.22, 12.18)	−1.93 (−56.18, 52.32)	4.59 (−8.76, 17.95)
Primero de Mayo	−4.80 (−16.44, 6.84)	−20.02 (−44.84, 4.79)	−2.54 (−7.85, 2.78)
Group PM+ Attendance			
No sessions	0.88 (−6.00, 7.76)	−0.75 (−15.70, 14.20)	2.41 (−5.60, 10.42)
One or more sessions	−0.21 (−3.71, 3.30)	−5.13 (−14.36, 4.10)	−1.52 (−4.84, 1.81)

**Table 3 ijerph-21-00527-t003:** RTC as a Mediator of Attendance.

**3a. Any Session Attendance (Binary)**					
	**Path A: Any** **Attendance →** **RTC**	**Path B: RTC →** **Outcome**	**Path C: Any Attendance →** **Outcome**	**Path C’: Any Attendance and RTC →** **Outcome**	
				**Any Attendance**	**RTC**
Depressive symptoms	3.80 (−0.65, 8.25)	0.10 (−0.06, 0.26)	1.26 (−2.03, 4.55)	0.45 (−3.04, 3.95)	0.10 (−0.06, 0.26)
Post-traumatic stress symptoms (log transformed)	3.80 (−0.65, 8.25)	0.02 (−0.02, 0.05)	0.22 (−0.59, 1.03)	0.16 (−0.70, 1.03)	0.01 (−0.02, 0.05)
Self-defined problems	3.80 (−0.65, 8.25)	0.08 (−0.09, 0.25)	2.69 (−0.85, 6.24)	3.42 (−0.30, 7.15)	0.05 (−0.12, 0.22)
**3b. Number of Sessions Attended (Continuous)**					
	**Path A: Number of Sessions →** **RTC**	**Path B: RTC →** **Outcome**	**Path C: Number of Sessions →** **Outcome**	**Path C’: Number of Sessions and RTC →** **Outcome**	
				**Number of Sessions**	**RTC**
Depressive symptoms	1.19 (0.27, 2.10)	0.10 (−0.06, 0.26)	0.28 (−0.42, 0.98)	0.10 (−0.64, 0.85)	0.10 (−0.07, 0.27)
Post-traumatic stress symptoms (log transformed)	1.19 (0.27, 2.10)	0.02 (−0.02, 0.05)	−0.01 (−0.18, 0.16)	−0.04 (−0.22, 0.15)	0.02 (−0.02, 0.06)
Self-defined problems	1.19 (0.27, 2.10)	0.08 (−0.09, 0.25)	0.19 (−0.57, 0.95)	0.26 (−0.55, 1.07)	0.07 (−0.11, 0.24)

## Data Availability

Please reach out to the corresponding author to request the dataset analyzed in this manuscript.

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
