# Peer review of "Comparing Mediators and Moderators of Mental Health Outcomes from the Implementation of Group Problem Management Plus (PM+) among Venezuelan Refugees and Migrants and Colombian Returnees in Northern Colombia"

_ijerph, 2024, doi:10.3390/ijerph21050527_

Round 1

Reviewer 1 Report

Comments and Suggestions for Authors

The manuscript identifies various mediators that show to affect the effectiveness of a task-shared, scalable psychological intervention, drawing on studies conducted among Venezuelan migrants and Colombian returnees in Northern Colombia. The research is important as this is one of the first studies looking into the mediators and moderators potentially affecting effectiveness of the intervention, which can have important policy implications. Overall, the manuscript is well-written, but I have a few questions and concerns that may help improve the quality of the paper.

My first concern is related to methodology. Namely, while the authors make sufficiently clear that the research on which the outcomes shared in the current study was not designed specifically for these questions, it is not clear to me what the study did consist of. For instance, the authors make reference to 'the parent study' several times, and refer to another article in which more details about the study can be found. I would like to ask the authors to show a bit more of this other study in this article too, to prevent readers from having to consult the other article to well understand this manuscript.

I would like to see a bit more description of the context that brought about Colombian returnees and Venezuelan migrants. Moreover, I missed a clear description of the three study sites and what that signifies. This seems particularly important as one of the findings points to particular/different findings in one of these sites.

It was not entirely clear to me whether the current study was conducted only with women or both men and women? I missed clear information about gender in the description and in table 1 (part of the interviewees were women, is made clear in the sampling description, but the other participants are simply called 'participants')

The qualitative data seems to consist of findings collected through FGD. The kind of questions asked in these FGD are mentioned in 2.2.4, but I would like to already get a sense of why or what kind of qualitative data was collected when this type of data is first mentioned, in 2.2.2.

Reading the section about analysis, I was wondering why authors make the choice to use both NVivo and Dedoose, both softwares to analyse qualitative data? 

Some questions in relation to the findings:

In 3.2 mention is made of a reduction in mental health outcomes. I wonder if this is was it meant, as I read it as 'worse’ mental health outcomes. 

Please check the translation in the quote in line 314. Currently, the first sentence does not make sense to me.

In the discussion section, it is suggested that the findings point to the idea that the intervention appeared less effective for individuals in less stable, more marginalized positions. However, two notable exceptions are shared as well: participants who experienced GBV and those who lived in one of the more marginalized (i.e. dangerous) of three sites. While the authors give possible reasons for this, I feel I miss some important qualitative information to interpret these surprising findings. More contextual information would be a way to resolve this concern.

In particular, the different findings in one of the sites seem puzzling to me, given that the information shared about the locality and people there, appear to suggest that the participants from that area are both more marginalized and less stable. What were the other two sites like, that make it plausible that in Primero de Mayo "there a critical void is filled" that is not filled in the other two sites? Was this the location where Colombian returnees were residing – thus suggesting that particularly citizenship status matters? (this would tie in with migration literature on wellbeing).

Similarly with GBV victims, it would be good to have more contextual information about these participants, to weigh/interpret these quantitative findings better – perhaps the authors can develop the argument better using the qualitative research findings?

In light of these findings, should the conclusion not be to strive towards a more specific understanding about the forms/types of instability and marginalization matter for in/effectiveness? Given the design of the study (geared towards other objectives) and size of the sample (relatively small), there may be various other specific forms of ‘less stable’ and ‘more marginalized’ characteristics that do not impede, but perhaps point to increased relevance, of this intervention?

Reviewer 2 Report

Comments and Suggestions for Authors

I enjoyed reading your effort very much, which I found generally clear and compelling. I have a very few suggestions. First, line 75 uses the phrase "looked at..." and I suspect you mean analyzed or examined? Or perhaps, investigated? Second, I did not understand the "waitlist" language in line 145. Why was such necessary exactly? Third, I did glean from your author statements who did what in this effort but I think it would help readers to have a paragraph contextualizing the roles of the various institutions and authors in the various elements of this research, including both its design and implementation. Fourth, I suggest you eliminate your use of passive voice on page 3 at lines 107, 109 and 130.

Comments on the Quality of English Language

I detected two or three extra word typos and some instances of passive voice that should be eliminated. This article was otherwise a pleasure to read.
